# Southern Ocean in-situ temperature trends over 25 years emerge from interannual variability

Matthis Auger [1,2✉], Rosemary Morrow[3], Elodie Kestenare [3], Jean-Baptiste Sallée[1] & Rebecca Cowley [4]

Despite playing a major role in global ocean heat storage, the Southern Ocean remains the most sparsely measured region of the global ocean. Here, a unique 25-year temperature time-series of the upper 800 m, repeated several times a year across the Southern Ocean, allows us to document the long-term change within water-masses and how it compares to the interannual variability. Three regions stand out as having strong trends that dominate over interannual variability: warming of the subantarctic waters (0.29 ± 0.09 °C per decade); cooling of the near-surface subpolar waters (−0.07 ± 0.04 °C per decade); and warming of the subsurface subpolar deep waters (0.04 ± 0.01 °C per decade). Although this subsurface warming of subpolar deep waters is small, it is the most robust long-term trend of our section, being in a region with weak interannual variability. This robust warming is associated with a large shoaling of the maximum temperature core in the subpolar deep water (39 ± 09 m per decade), which has been significantly underestimated by a factor of 3 to 10 in past studies. We find temperature changes of comparable magnitude to those reported in Amundsen–Bellingshausen Seas, which calls for a reconsideration of current ocean changes with important consequences for our understanding of future Antarctic ice-sheet mass loss.

[1] Sorbonne Université, CNRS, LOCEAN, Paris, France. [2] CNES, Toulouse, France. [3] LEGOS, CNRS/IRD/CNES/University of Toulouse III, Toulouse, France. [4] CSIRO Marine and Atmospheric Research, Hobart, Tasmania, Australia. ✉email: matthis.auger@locean-ipsl.upmc.fr

The Southern Ocean has been rapidly changing over the past decades with widespread consequences for the global climate. It has stored an outsized amount of heat associated with climate change that has been extracted from the atmosphere and stored in its subsurface water-masses[1,2]. The Southern Ocean alone has stored 35–43% of the global upper 2000 m ocean heat gain from 1970 to 2017, and an even greater proportion in recent years, with an estimate of 45–62% from 2005 to 2017[2]. This heat storage, as well as concomitant change in its vertical stability due to change in surface salinity[3–5], translates into significant warming of subsurface water-masses[6]. The overall water-mass warming since 1970 is composed of significant warming north of, and within, the eastward flowing Antarctic Circumpolar Current[7–9] (ACC), and slight cooling observed in the surface subpolar waters[10]. Some regions show slight warming and uplifting of the subpolar Upper Circumpolar Deep Waters (that lie directly offshore the Antarctic continental shelf), threatening to invade onto the continental shelves with drastic potential consequences for the melt of Antarctic Ice Shelves and subsequent global sea level rise[11].

Despite those emerging results, there are inherent limitations in our past and current observation system that pose a strong limitation in our confidence of any of these climate-scale changes that occurred in the Southern Ocean[12,13]. For most changes in the Southern Hemisphere, it remains unclear whether the natural and interannual variability can cause the observed change or overwhelms the forced response[13]. A recent study based on numerical simulations suggests that warming north of the Antarctic Circumpolar Current is largely human induced and overwhelms the natural variability[14]. But this remains one study using one single climate model, and our limited confidence in the representation of subpolar Southern Ocean processes in climate models drastically hampers our confidence at higher latitude[2]. Observations are needed, more than in any other region, to shed more light on long-term ocean trends and understand how they compare to natural and interannual variability.

In this paper, we unlock these limitations by presenting an observation dataset of the most frequently repeated and longest time-series of a temperature section across the Southern Ocean in the upper 800 m, from its northern boundary to Antarctica. The temperature section, referred to as Section IX28, is the longest of the three long-term high-resolution repeat upper ocean XBT temperature monitoring lines that have made observations of the seasonal heating cycle across the Southern Ocean[15]. IX28 has been repeated several times a year since 1992 at 140°E, from Hobart, Tasmania to Antarctica (Fig. 1a), providing us with a unique 25-year temperature time-series to robustly estimate summer temperature changes consistently across an entire meridional section, and document from observations how temperature changes compare to typical interannual variability.

## Results

**25-year Climatological state and long-term change.** Based on the 148 repeats of the same section, we construct a summer temperature climatological mean over the 25 years (since November 1992), which shows the main Southern Ocean water-masses and the fingerprints of the main fronts associated with the Antarctic Circumpolar Current (Fig. 1b; see "Methods"). The warmest water-masses on the section, the Subtropical Water (STW) and SubAntarctic Mode Water (SAMW) are located in the northern part of the transects. Their southern extent is limited by the Subtropical Front (11 °C at 150 m[16]) and the Subantarctic Front (strongest temperature gradient between 3 and 8 °C at 300 m depth[17]), respectively. SAMW is found down to 600 m depth, beneath the summer mixed layer, consistent with previous studies[18,19]. Antarctic Surface Waters (AASW) are located in the

upper 250 meters of the Southern Ocean and south of the Polar Front (most northern extent of the subsurface 2 °C water[20]). AASWs are composed of a remnant subsurface tongue of cold water produced in winter[21,22] (Winter Water), and warmer surface waters produced in summer[23,24]. Below the Winter Water tongue lies the less-dense Upper Circumpolar Deep Water (UCDW), then the denser Lower Circumpolar Deep Water (LCDW), that rises beneath the WW layer south of the Antarctic Divergence around 63°S. These Circumpolar Deep Waters are advected at depth around the Southern Ocean, and partly originate from North Atlantic Deep Waters[25].

We are interested in how this temperature structure is changing over time on a multi-decadal timescale. Over the past decades, the temperature has been warming overall across the section, but with a structure showing marked patterns, which are related to the different water-masses of the region. The largest warming reaching 0.4–0.8 °C per decade is observed on the northern end of the section, north and within the ACC (region A in Fig. 2b) in the subtropical waters and subantarctic Mode Waters. In contrast, on the southern end of the section, a cooling trend of 0.1–0.3 °C per decade is observed in the coolest water-mass of the region (region B in Fig. 2b), extending from the surface to about 200 m, in a region where the interannual variability has similar magnitude. Hints of cooling trends are also apparent in the surface layer further north, but the trends are dominated by interannual variability north of ~61°S in the surface layer. Deeper in the water column, the Upper Circumpolar Deep Water layer (region C in Fig. 2b) shows subtle warming trends of around 0.05 °C per decade from 62.5°S to 52°S, but here, the interannual variability is weak.

The temperature change structure shown across the section concurs well with past studies that have investigated long-term temperature trends in the Southern Ocean (ref. [6], and references therein). Here, we however bring an important step forward in our understanding of past changes by showing that Southern Ocean water-mass temperature trends is robust over a 25-year period. But more importantly, we are able to estimate the typical interannual variability (referred to here as noise) to better interpret the observed trends over a 25-year period (referred to here as signal; see Methods). In other words, from observations in the Southern Ocean, we are able to estimate whether the signal of temperature change has emerged above the interannual variability noise. A latitude-vertical section of this trend signal-to-noise ratio is shown in Fig. 2c. The three regions highlighted above clearly stand out, experiencing temperature changes that emerge above the background interannual variability over the past 25 years. Counter-intuitively, it is in the Upper Circumpolar Deep Water layer, where the long-term change amplitude is the lowest of the section, that the signal-to-noise ratio is the largest because interannual variability is actually very weak. This clearly pinpoints that, while subtle, the observed temperature increase in the Upper Circumpolar Deep Water represents a radical deviation from its mean state. In other water-masses with a more recent surface connection, the 25-year trends are weaker compared to the typical interannual variability. A signal-to-noise ratio lower than one does not mean trends are insignificant; rather it remains unclear whether the measured long-term change reflects a robust change departing from its typical interannual variability. A robust long-term trend might be hidden behind a low signal-to-noise ratio, but one would have to accumulate more years of repeat observations to observe its emergence above the interannual noise.

**Water-mass temperature time-series and forcing.** We next compute time-series and associated trends, averaged over the three regions identified above where trends overcome both their

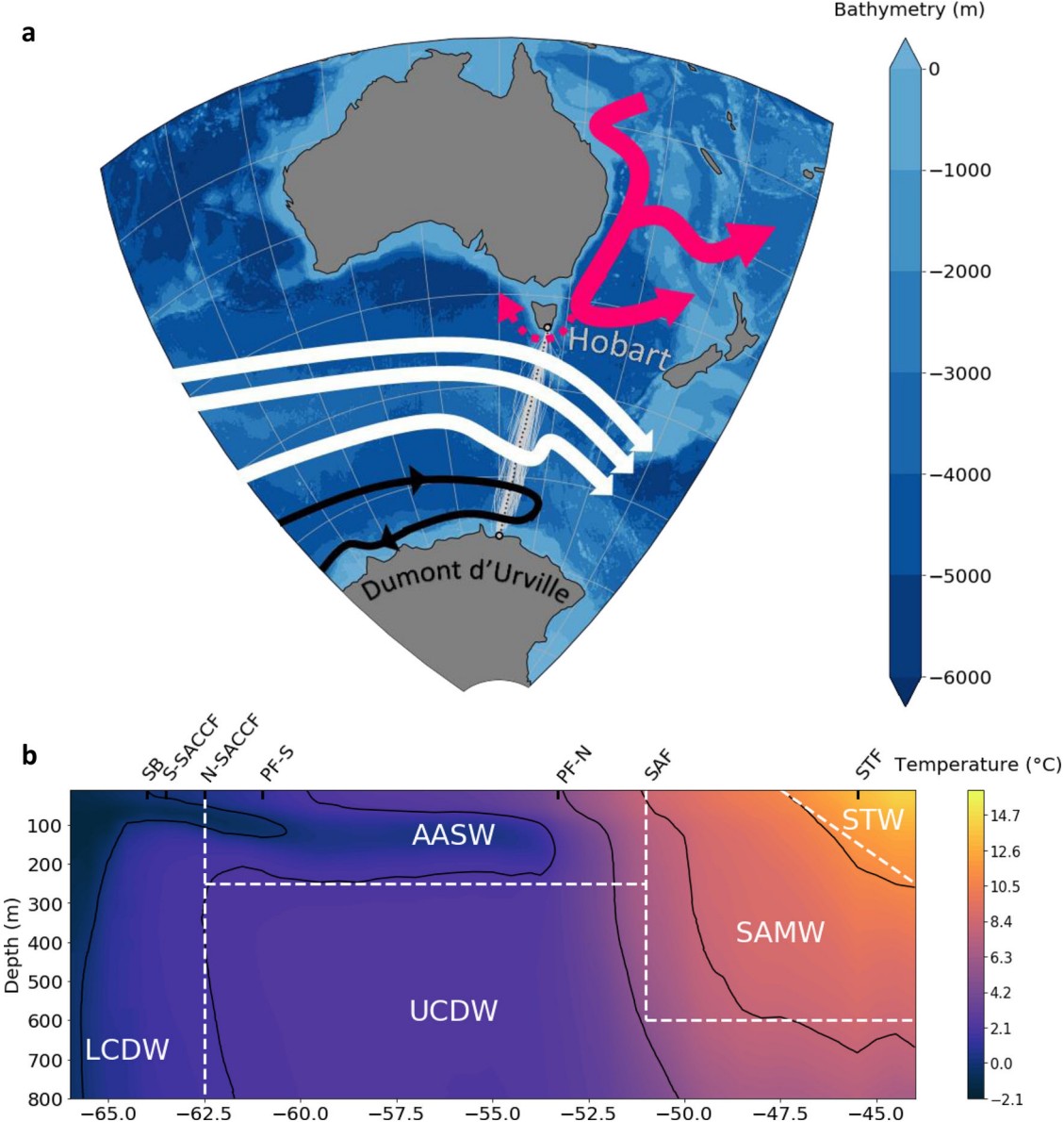

**Fig. 1 SURVOSTRAL program transects and summer mean temperature section. a** SURVOSTRAL observations over 25 years between Hobart and Dumont D'Urville (DDU), and bathymetry of the region. The mean trajectory is in dashed black. Data used in this study are in gray. A schematic circulation is represented. White, black, and red arrows are respectively the Antarctic Circumpolar Current, the Antarctic Slope Current and Australian-Antarctic Basin gyre, and the East Australian Current. **b** 25-year average of the summer (NDJF) temperature sections. Average position of the fronts (SB: Southern Boundary, S-SACCF: Southern Branch of the Southern Antarctic Circumpolar Current Front, N-SACCF: Northern Branch of the Southern Antarctic Circumpolar Current Front, PF-S and PF-N are the Southern and Northern branches of the Polar Front, SAF: SubAntarctic Front, STF: SubTropical Front) and principal water-masses positions are indicated (LCDW: Lower Circumpolar Deep Water, UCDW: Upper Circumpolar Deep Water, AASW: Antarctic Surface Water, SAMW: SubAntarctic Modal Water, STW: SubTropical Water). Black contours show the mean isotherms.

standard error, and the typical interannual variability: in the subantarctic and subtropical region north of 52.5°S (region A); in the near-surface subpolar region, in the upper 200 m, south of 61°S (region B); and in the subsurface Upper Circumpolar Deep Water, deeper than 250 m, and between 62.5°S-55°S (region C).

When averaged over the entire Subantarctic and Subtropical Mode Water region (region A), the temperature has increased significantly by 0.29 ± 0.09 °C per decade, with a 25-year signal to noise ratio of 2.40, indicating a trend much greater than the estimated interannual noise (Fig. 3a). Locally the trend can be as high as 0.8 °C per decade (Fig. 2b), with the strongest warming organized in deep-reaching localized vertical bands. These structures may be related to more prevalent warm-core eddies

or small meanders towards the end of the time series. We note that the computed warming is similar when analyzed in streamwise coordinates following altimetric-derived meanders or in geographical coordinates[26].

Based on a shorter 13-yr time-series, Morrow et al.[27] proposed that this warming was due to the southward movement of both the STF and the SAF, reflecting the consensus when the study was published that ACC fronts were shifting southward. After a decade of scientific debate, a new consensus emerges that on a circumpolar average, the SAF has been shown to be stable and not moving meridionally in the last decades[2,28] and that the warming might instead be due to increased heat uptake from the ocean surface[19,29]. While the warming trend is relatively constant

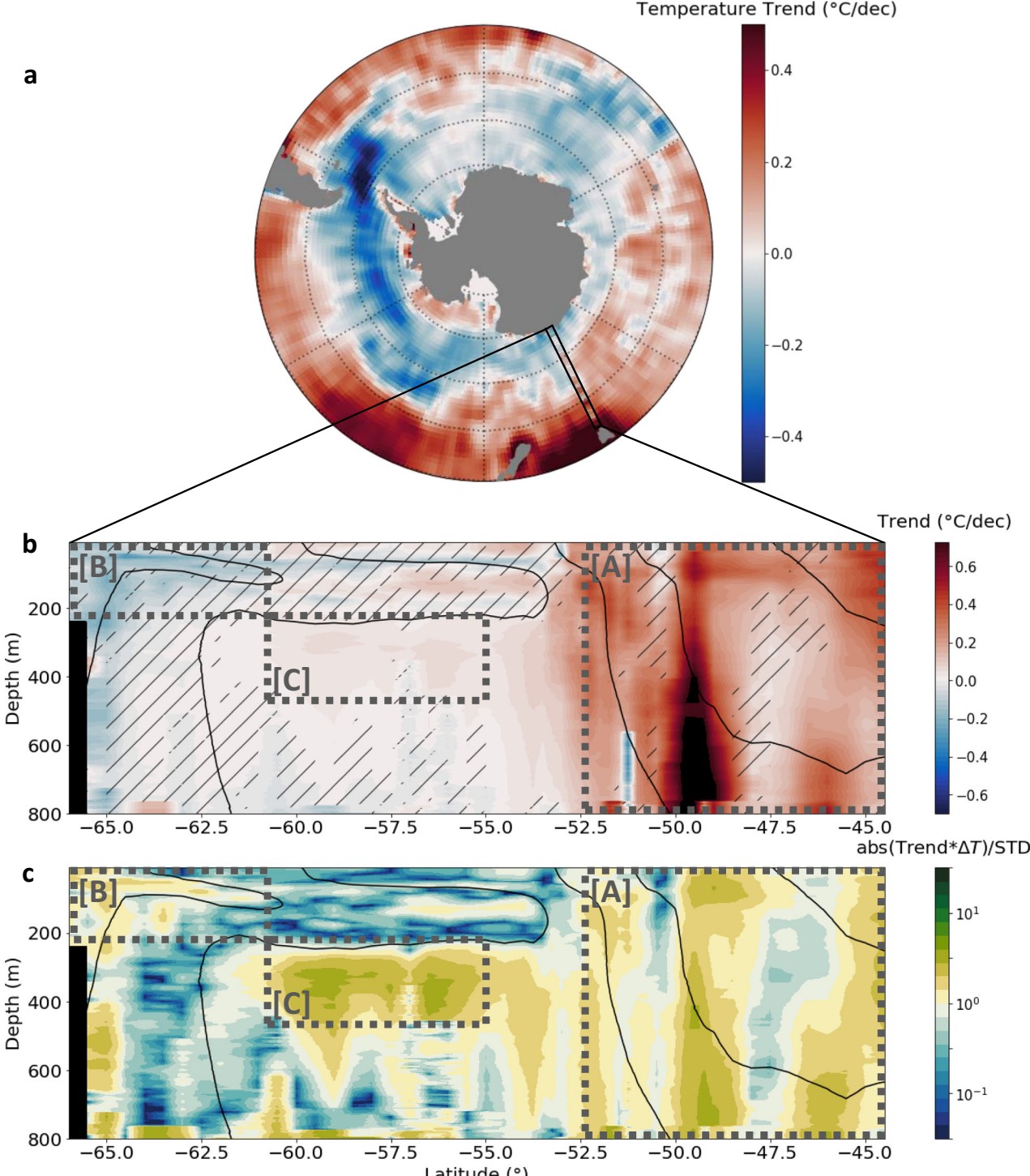

**Fig. 2 Temperature trend section and its ratio with interannual variability. a** Summer Reynolds SST Trends (°C/dec) from 1993 to 2017 (NDJF). Black box indicated the region of SURVOSTRAL transects. **b** Temperature trends (°C/dec) from SURVOSTRAL XBT data. Hatched data represent zones where abs(Trends*ΔT)/STD < 1, ΔT being the length of the record; i.e., where the trends are smaller than the interannual variability over the 25 years of measurements. **c** Ratio between the trend signal and interannual variability. Position of zones [A], [B], and [C] discussed in this study is represented by the dotted boxes in (**b**, **c**).

over the 25-year period, there are periods of distinct cooling, for example, in 1996 and 2005, and stronger warming in 2001–2002 and in 2014–2016. Similar interannual variability is also evident in the sea-surface temperature fields, with a correlation of 0.63 between SST and the Region A temperature time series, and a slightly lower 25-yr trend of 0.15 ± 0.09 °C per decade, consistent

with the trend distribution within the zone (Fig. 2b). Part of the observed interannual variability might be due to intermittent incursions of subtropical waters carried by the Tasman Sea extension south of Tasmania, impacting the extent of STW, as well as local eddy activity around the SAF[30–32] (See Supplementary Fig. 1).

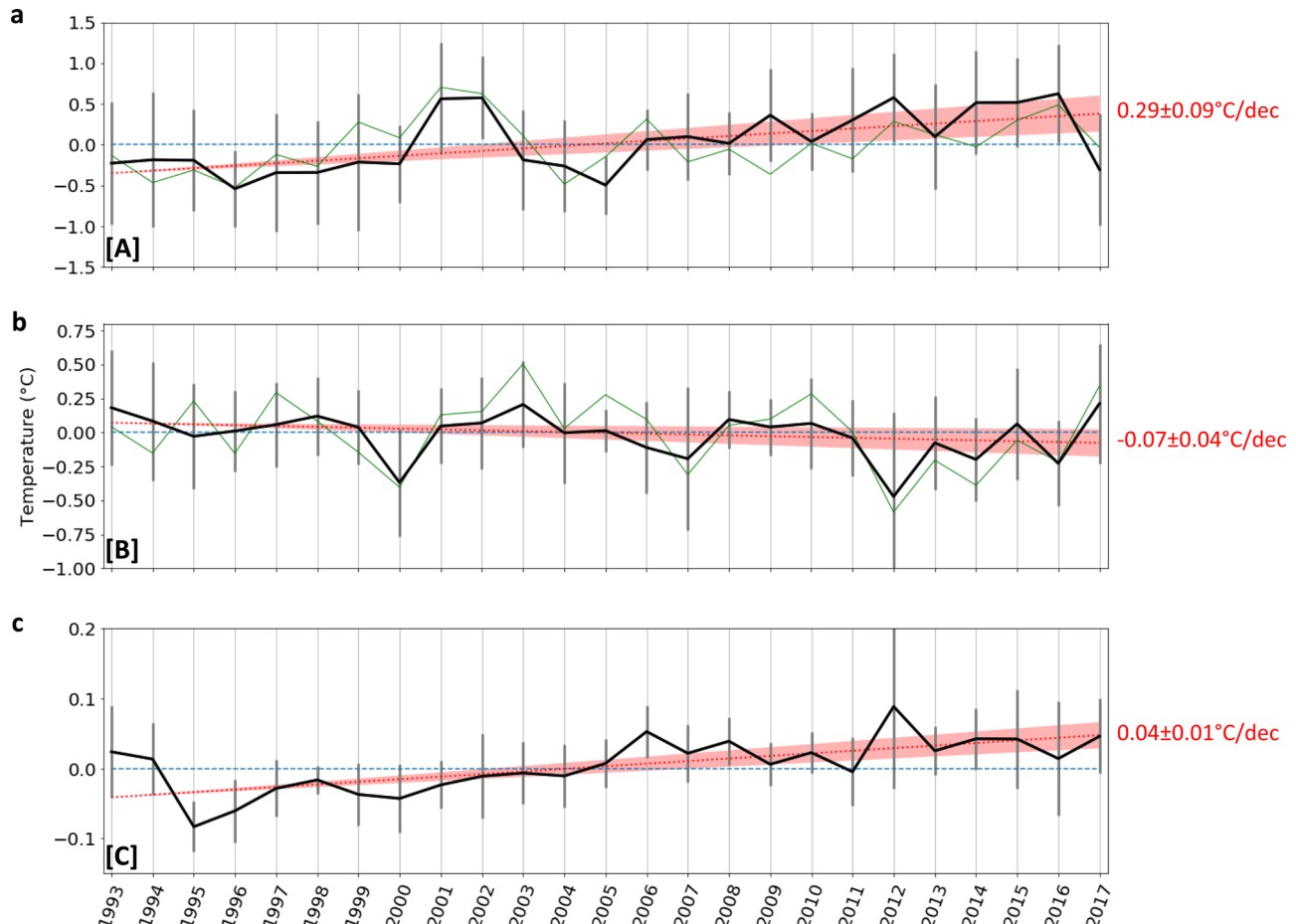

**Fig. 3 Temperature anomalies time-series and trend per sector. a–c** Show the evolution (black line) and trend (red line) of the temperature anomalies within zones [A], [B], and [C], respectively. Green line is the NDJF SST Reynolds anomalies interpolated onto the SURVOSTRAL line for each zone. Errors bars are the standard deviation of the mean anomalies for each grid point within the zone.

The overall cooling in the surface subpolar waters close to Antarctica, from the surface to 200 m and from 66°S to 61°S (region B), has a non-significant trend of −0.07 ± 0.04 °C per decade (Fig. 3b, *p*-value 0.07), with a signal-to-noise ratio of 1.16. The cooling appears mostly associated with the coolest waters in the regions (Fig. 2b); Figs. 1b and 2b, c both show water-mass cooler than 0 °C as standing out at the southern edge of our section, with consistent long-term change. When subjectively isolating only data points cooler than 0 °C, the cooling is significant and slightly more marked (−0.09 ± 0.05 °C per decade, signal-to-noise ratio of 1.49; Fig. 4a). This cooling of subpolar waters is also accompanied by a freshening of the surface waters over the same period, as well as an increase in sea-ice cover[5]. Region B has a lower signal-to-noise, and the interannual variability in temperature, SSS and sea-ice is impacted by local coastal circulation changes and increased ice flow from 2011 onwards, following the Mertz Glacier calving just upstream[33–35]. Such high-latitude cooling over the upper 200 m in region B is also consistent with local sea surface cooling observed from satellite SST observations (Fig. 3b, correlation *r* = 0.80), and more generally with the surface cooling of a large part of the Southern Ocean that have been observed from observations in the subpolar waters over the past three decades[10,13,36]. This cooling has been explained by the increased stratification associated with freshening of the surface layer which would tend to reduce mixing with the slightly warmer underlying Lower and Upper Circumpolar Deep Water[4,10,37–39]. Indeed, a trend in surface water freshening

has been observed over the same period near 140°E[5]. This has been linked to increased sea-ice cover, particularly after the Mertz Glacier calving in 2010 and enhanced by a large-scale northward shift of the zero-zonal wind position from 1999 onwards, that increased the Ekman-driven sea-ice convergence near the coast[5].

Interestingly the winter water tongue extending further north does not show a similar cooling. Small pockets of cooling exist but the WW trend signals are dominated by interannual variability (0.22 signal to noise ratio). Even when focusing only on the temperature of the core of the Winter Water layer, defined as the layer with temperature colder than 2 °C between 55°S and 61.5°S, the large interannual variations overwhelm any long-term change, with peak-to peak temperature ranging from 0.40 to 0.65 °C (Fig. 4b). These temperature variations within the Winter Water core are positively correlated (*r* = 0.70) with the sea surface temperature of the previous winter further upstream in the subpolar Australian-Antarctic basin (120–145°E; 57–61°S) (Fig. 4b), where the Winter Waters were modified at the surface (see Supplementary Note 1).

The upper layer of the Upper Circumpolar Deep Water from 61°S to 55°S, and over 250–450 m depth (region C) exhibits a small but significant overall warming trend of 0.04 ± 0.01 °C per decade (significant, Fig. 3c), associated with a high signal to noise ratio of 2.58. Consistently, the time-series show relatively weak interannual variability, but a steady warming of the layer. The maximum temperature increase sits directly below the seasonally variable surface layer, in the upper and warmer part of the

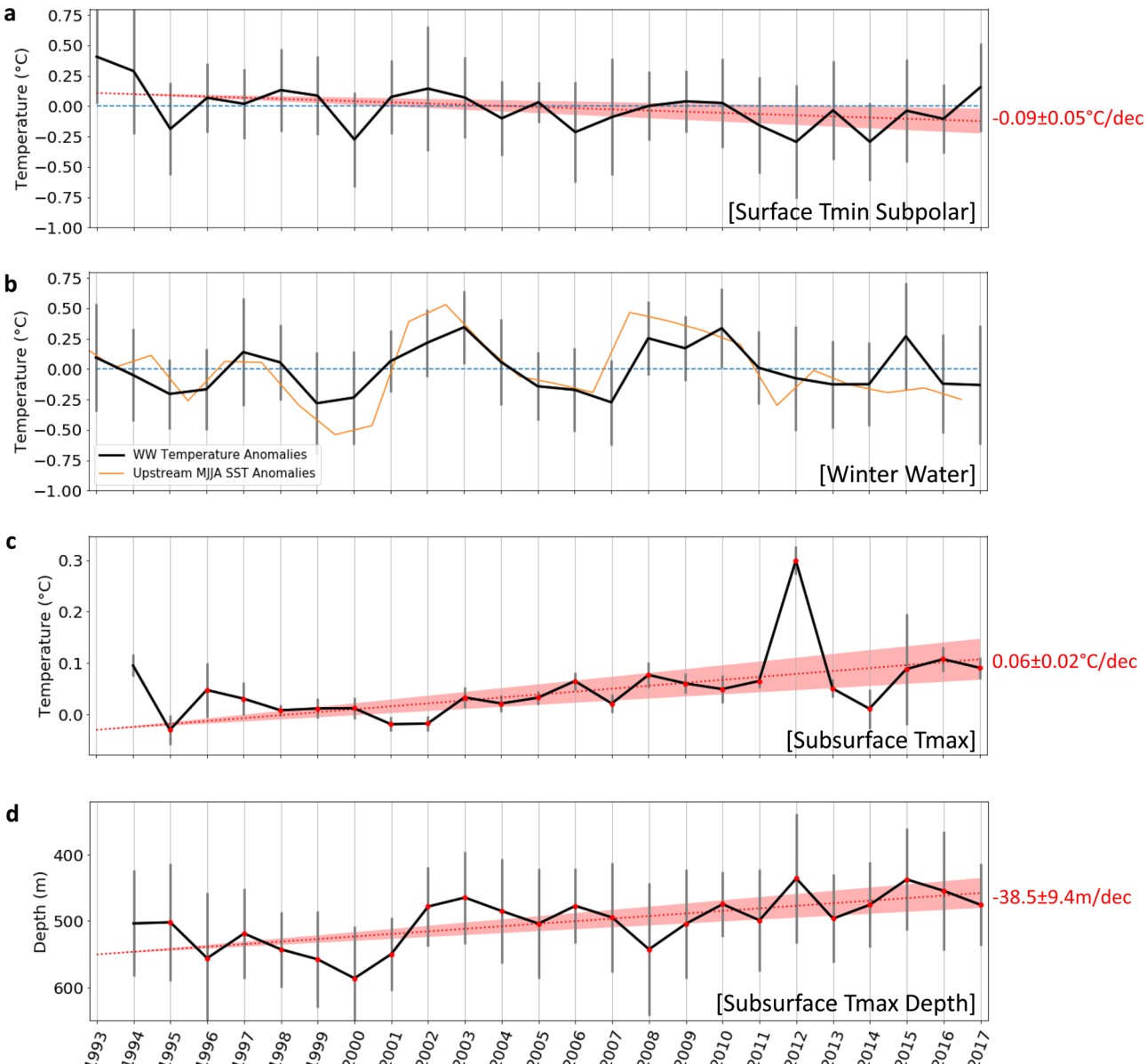

**Fig. 4 Time-series of temperature and characteristics of specific water mases. a** Zone [B] anomalies, restricted to the gridpoints where the 25 years-mean temperature transect <0. **b** Black line is the WW temperature anomalies from SURVOSTRAL XBTs between 54 and 61.5°S, restricted to the Tmin gridpoints where the 25 year-mean temperature transect is less than 2 °C. Yellow line is the MJJA SST anomalies upstream of SURVOSTRAL WW, between (120–145°E and 57–61°S). **c** CDW maximum temperature evolution (see "Methods"). Red dots are the years the linear trend is computed on; i.e., years when there is at least 2 months with data on average for each grid point for NDJF months. **d** CDW maximum temperature depth (see "Methods"). Errors bars are the standard deviation of the mean temperature anomalies (depth for panel **d**) for each grid point within the zone.

water-mass around 300–550 m (Fig. 2b). When the temperature time-series is computed in this core of temperature maximum, the warming trend is even greater, reaching 0.06 ± 0.02 °C per decade, with a signal to noise ratio of 2.44 (when excluding 2012 which appears as a clear warm outlier, the trend is the same as in the full region C, being 0.04 °C ± 0.01 °C per decade, but with a higher signal to noise ratio of 3.79). Previous authors have suggested the warming of the Upper Circumpolar Deep Water might be driven by increased stratification at the base of the Winter Water layer due to freshening, which would reduce mixing between the two layers and heat removal from the Upper Circumpolar Deep Water to the atmosphere[10,40,41]. Since we have only temperature profiles, the role of the salinity stratification cannot be verified directly. However, in accordance with this hypothesis, we observe larger warming in the upper part of the

layer, directly underlying a near-surface water mostly affected by interannual variability (Fig. 4b) but with a few hints of local cooling (Fig. 2b). In addition to the warming of Upper Circumpolar Deep Water, the depth of the core of maximum temperature is observed to shoal at a significant rate of 39 ± 9 m per decade (Fig. 4d), three to ten times higher than previously reported (5–10 m per decade[11]), and within the error envelope of the rate observed in West Antarctica (50 ± 18 m per decade[11]). The cause of the shoaling of the maximum temperature layer remains unclear. It could be related to long-term changes in Ekman pumping[11], but using the atmospheric reanalysis ERA-5, we find only a very subtle long-term trend in local upward Ekman pumping, which is not statistically significant. Other potential mechanisms, e.g., associated with turbulence-driven shoaling of the surface layer, remain to be tested in a future study.

## Discussion

Our findings carry important implications for our understanding of Southern Ocean temperature change, a region of the world that remains poorly observed and understood, though with a pivotal role in global climate. Using a unique observation time-series repeated several-times per year over the past 25 years across the Southern Ocean, we document the temperature trend over the upper 800 m, and shed light on three main regions where the temperature change dominates over typical interannual variability. Interestingly, only the subtropical region, north of the Antarctic Circumpolar Current (region A) has been shown to be associated with a human-induced forced signal that emerges over natural variability[14], though recent work suggests that forced warming in the sub-surface subpolar ocean does emerge over natural variability by the end of the 20th century or early decade of the 21st[42]. We note that these studies are based on climate models with significant limitations in their representation of the Southern Ocean[2], hence it is important to provide robust observational targets for future improvement.

The repeat meridional temperature sections used in this study cross a Southern Ocean region of inter-ocean exchange, where waters from the Pacific can flow south of Tasmania into the Indian Ocean[30,43]. The northern part of the IX28 section exhibits strong interannual variations in the temperature data, impacted by ENSO/SAM climate modes and eddy movements across 140°E[44]. Despite this, our 25-year trend calculations have a strong signal-to-noise, with the upper ocean warming trend exceeding the interannual variations. The warming of $0.29 \pm 0.09$ °C per decade north of the ACC is in accordance with previous studies[7–9] and with other parts of the Southern Ocean[45]; Southern Ocean circulation being essentially zonal, subsurface trends are expected to be zonally consistent all-around Antarctica. Close to the Antarctic continent, during the austral summer heating cycle, our temperature profiles confirm that the widespread surface cooling around Antarctica observed with satellite SST data extends to around 200 m depth at 140°E.

One of the most important results of our study is the large warming and shoaling of the subsurface temperature maximum in the subpolar Southern Ocean, in the Upper Circumpolar Deep Water. This water-mass sits directly below the surface layer and mostly flows eastward, feeding the Pacific basin, where major increase of basal melt has long been identified further downstream in the Amundsen–Bellingshausen sector[46]. In addition, we note that some of the water-masses at the southern end of the section, though probably south of the maximum Upper Circumpolar Deep Water warming we observe, might be part of a cyclonic Australian-Antarctic gyre[47], with direct influence on the Wilkes basins that has recently been shown to be associated with important mass loss of many glaciers of this region[46,48–50]. Our 25-year study confirms two major threats (significant warming and shoaling of Upper Circumpolar Deep Water) that may enhance the ice-shelf melting downstream, with potential dramatic impacts for future global sea-level. Both of these changes that we observed at 140°E have been substantially underestimated in this part of the Southern Ocean until now and must imperatively be taken into account in future ice-sheet modeling predictions[51], and more generally when developing future climate change narratives. Our observational study provides a basis for validating such models and contributing toward these developments.

## Methods

**SURVOSTRAL Program**. The dataset of temperature used in this study consists of 25 years (November 1992 to February 2017) of XBT profiles on a section from Hobart (Tasmania, 42.9°S, 147.3°E) to Dumont d'Urville (Adelie Land, 66.6°S, 140.0°E), as part of the SURVOSTRAL project (Fig. 1a, https://doi.org/10.18142/

172). Measurements are taken from the French Antarctic resupply vessel *L'Astrolabe*, with about six transects per year between late October and early March. Depending on ice and weather conditions, XBT measurements are sampled every 35 km, with 18 km sampling across the energetic polar frontal region. Temperature profiles extend down to 900 meters depth with a vertical resolution of about 0.7 meters. The XBT temperature profile accuracy is +0.1 °C. XBT profiles over the entire series have been corrected for temperature and depth biases depending on the probe type, following refs. [52,53]. Corrected XBT measurements are available here: http://thredds.aodn.org.au/thredds/catalog/IMOS/SOOP/SOOP-XBT/PRODUCTS/BiasCorrectedData_ChengEtAl_2014/Line_IX28_Dumont-d-Urville-Hobart/catalog.html.

**Gridding process**. In order to compute anomalies and trends, 10238 XBT profiles are interpolated onto a regular line from North to South, following the mean path of the Astrolabe's transect, with 0.5° resolution in latitude (increasing to 0.25° in the polar frontal zone from 49 to 54°S), with 2 m depth resolution down to 800 m depth. Results are robust when changing the vertical resolution and interpolation type. XBT profiles sampled further than 3° in longitude from the mean path of the Astrolabe are removed from the analysis. In the following sections, we will discuss three types of products on this regular grid.

1. Climatological monthly mean temperature sections are calculated for each month during the austral summer ONDJFM period and averaged over 25 years. Since the sections are not evenly distributed within a given month, each monthly temperature section is assigned to the median sampling day of all profiles in the month. These values are then linearly interpolated onto daily values before calculating temperature anomalies.
2. Temperature anomaly profiles are constructed by subtracting the corresponding climatological daily value at each latitude and depth from each measurement. These anomalies allow us to construct a gridded section of interannual temperature anomalies and the temperature anomaly trends for the 25-year observation period.
3. Annual austral summer (NDJF) mean temperature sections are constructed for each year from 1993 to 2017 (Supplementary Fig. 2). This product is only used in this study to locate the CDW temperature maximum zone.

The data distribution and the main data processing techniques for these three products are provided in the supplementary information. The monthly mean temperature sections from October to March (Supplementary Fig. 3) calculated from the 25-year time series are consistent with those calculated by ref. [23] based on 8-years of SURVOSTRAL data. This highlights that the seasonal warming cycle is quite stable in this region on a long-term average. The water-masses with the strongest seasonal changes are at the surface: the Antarctic Surface Waters (AASW) south of the Polar Front show the largest monthly mean variations over the summer warming cycle with coolest waters observed in sampled months closest to winter, late Oct–Nov. In the north of the section, there is a seasonal southward and deepening expansion of Subtropical waters throughout the summer season. We note that even if measurements are sampled only in summertime, computed trends can be considered as annual trends. Indeed, the main seasonal variations are in the surface layer, and XBT temperature profiles' surface values are consistent with satellite SST values. Finally, SST trends computed on NDJF months are coherent with SST trends computed on full year. This shows that for the surface layer, there are no wintertime trends that are counteracting the summer trends, and observed trends are consistent for the whole year for the full time series.

**Trend section and zone trends**. The temperature trend latitude-depth section is constructed by computing a linear trend using the anomalies available at each grid point. Each profile is associated with one latitude in the grid and is interpolated onto the depth grid. No interpolation was made in latitude to avoid interpolation of anomalies over large data gaps (e.g., during storms), so trends are robust to varying data distribution. The yearly anomalies are weighted by 1/std of all of the anomalies obtained during the corresponding season. The number of measurements used to compute the 25-year trends for each grid point is represented on Supplementary Fig. 4. Each grid point is sampled by between 3 and 10 profiles per year. With an XBT accuracy of 0.1 °C, it translates into a standard error from ~0.03–0.06 °C, allowing us to resolve changes over 25 years of 0.001–0.002 °C per year, or 0.01–0.02 °C per decade. This value is lower when computing trends over larger regions A, B, and C. Surface trends are consistent with SST Reynolds[54] product trends on summer NDJF periods (Fig. 2a, $r = 0.70$), and SST Reynolds[54] full year trends ($r = 0.70$). Trends averaged over zones [A], [B], and [C] are computed in the same way, but all anomalies available in each zone are averaged for each season. The trend significance is computed using a Mann-Kendall test. Trends with $p$-value lower than 0.05 are considered significant, and their confidence interval is computed as their standard error.

CDW maximum temperature values and their depths are computed by selecting the warmest 10% temperature grid points on each austral summer temperature section within zone [C]. The mean depth of these selected grid points is then the depth of maximum CDW temperature, and the mean anomalies of these selected grid points gives the evolution of the temperature maximum. CDW maximum temperature value and depth trend is computed only on the years when there is at least 2 out of 4 months with measurements on average for the summer NDJF mean

for all the subset grid points. Missing data in 1993 occurs since data is available in less than 10% of the grid subset.

**Trend (signal) to interannual variability (noise) ratio**. The amplitude of the trend compared to the strength of the interannual variability is evaluated for each zone and grid point, by computing the signal to noise ratio. Our signal is the temperature evolution following the linear trend over the 25 years, and our noise is the standard deviation of the error between the trend and the measured temperature:

If $T$ is the temperature evolution throughout the ny = 25 years, and $ax + b$ its linear regression, the signal to noise ratio $S$ is computed as:

$$S = \frac{\text{ny} * a}{\text{STD}(ax - T)} \tag{1}$$

$S$ represents the ratio between the trend and the interannual signal: if $S > 1$, the trend signal is dominant compared to the interannual variation.

**External data**. We use NOAA monthly optimum interpolation (OI) satellite and in-situ[54] surface temperature data to verify the consistency of our XBT observations to surface changes in temperature.

ECMWF ERA5 monthly surface turbulent wind stress product is used to investigate the effect of the wind on the temperature trends and variations (DOI: 10.24381/cds.f17050d7).

## Data availability

Corrected XBT measurements are available here: http://thredds.aodn.org.au/thredds/catalog/IMOS/SOOP/SOOP-XBT/PRODUCTS/BiasCorrectedData_ChengEtAl_2014/Line_IX28_Dumont-d-Urville-Hobart/catalog.html. The datasets generated during the current study are available with the DOI: 10.6096/11.

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

## Acknowledgements

XBT temperature data were obtained with the support of the French Institut Polaire Emile Victor (IPEV), the LEGOS, IMOS (Australia's Integrate Marine Observing System) and the CSIRO Marine and Atmospheric Research. We wish to thank the crew of the Astrolabe and the many volunteers who helped with these observations over 25 years. XBT data processing was performed by Rebecca Cowley of CSIRO, and the XBT technical assistance by Matt Sherlock and Craig Hanstein, CSIRO. This project is supported through funding from the Earth Systems and Climate Change Hub of the Australian Government's National Environmental Science Program. This analysis was financed by the French TOSCA program. M.A. and J.B.S. have received funding from the European Union's Horizon 2020 research and innovation program under grant agreement N° 821001. M.A. was funded through a CNES/CLS scholarship. Data were sourced from Australia's Integrated Marine Observing System (IMOS)—IMOS is enabled by the National Collaborative Research Infrastructure strategy (NCRIS). It is operated by a consortium of institutions as an unincorporated joint venture, with the University of Tasmania as Lead Agent.

## Author contributions

All analysis for this paper was performed by M.A. and supervised by R.M., E.K., and J.-B.S. The study was designed by R.M. Observational data were validated and calibrated by R.C. and E.K. All authors contributed to interpreting the results and writing the manuscript.

## Competing interests

The authors declare no competing interests.
