## [Peer Review File · Nature Communications]

REVIEWER COMMENTS

Reviewer #1 (Remarks to the Author):

This paper uses a 25-year time series of subsurface temperature measurements in the Australian sector of the Southern Ocean to distinguish long-term trends from year-to-year variability. The paper is very well written and the science is sound and represents a substantial contribution to the field. In particular, the results are important since we have very few long observational time series in the Southern Ocean and so much of our understanding of long-term changes in the region stem from models or sparse observational measurements of shorter duration. The unique aspect of this 25-year XBT transect is that the temperature profile data are collected along a near-repeat section and so it enables the analysis to be able to tease out temporal from spatial variability. I find that the paper should be suitable for publication in Nature Communications after addressing some of my concerns below.

Major Issues:

1. The linear trends could use a more rigorous error analysis rather than just comparing to the standard error of the variability.
2. The authors use a 25-year time series but it mainly consists of measurements from the summer season (NDJF) that are then used to form estimates of annual temperature trends. The paper convincingly shows the long-term trends from the constructed "annual" trends in some regions lie above the year to year variability. But this assumes that there are no counteracting trends that might occur in winter time measurements. This needs to be recognized and more fully discussed in the text.
3. Discussion of findings to other regions of the Southern Ocean?

Minor Issues:

1. Abstract: lines 31-33. Why this result is "counter-intuitive" is not clear here. Obviously, it becomes clear later in the paper why this is an unexpected result but the abstract should stand alone. The sentence needs a qualifier to help the reader understand why the result is counter-intuitive. In addition, the clause "largest change of the section regarding interannual variability" is also a bit ambiguous. As I read it in your analysis, despite showing only small long-term trends, Region C has one of the most robust long-term trend along the section as the trend rises well above the year-to-year variability.
2. Abstract: line 36. West and East are a bit ambiguous when referring to a polar region. I realize that there is the Western Antarctic Peninsula, but the region that I think you are referring to is in the Bellingshausen Sea. Perhaps instead of West Antarctica you could specifically state that you are referring to the Bellingshausen Sea downstream from your study area.
3. Lines 49-55. This is a long sentence that needs to be broken up for readability.
4. Lines 96-98: Figure 1 also distinguished LCDW at the far south along the transect. Should this also be discussed in this paragraph?
5. Lines 96: As I have indicated above, a more rigorous error analysis probably is needed to help discern which trends are statistically significant. Although helpful, a standard error is not sufficient to statistically determine whether a linear trend is significant or not. At the very least the uncertainty of a linear trend at 95% confidence is related to twice the standard error. I suggest the authors use more standard statistical tests, such as t-tests that account for effective degrees of freedom to test for their significance of the linear trends.
6. Line 118: Does a 25 year time series really resolve decadal time scales?
7. Line 150: "Deep-reaching localized cells". Rather than these features being called "cells" which suggests a circulation feature, could they be associated with shifting preferences in the location of the main frontal features that vary from cruise to cruise and year to year? This also points to the issue that averaging the temperature data by latitude necessarily convolves variability in the fronts into this geographical space. Ideally this averaging might be performed for different frontal regions. This

probably needs to also be addressed in the paper.

8. Line 159: What is the 0.64 correlation referring to? Between what time series?
9. Line 167: Should this refer to Figure 2c here rather than 3b?
10. Line 175: Is this the correct Figure citation? What is the correlation between?
11. Line 180-182: I wanted a bit more elaboration here about what the specific relationship is. What are the mechanisms that are being invoked for the freshening?
12. Lines 195-197: Reference Figure 3c here.
13. Line 204: I think you can delete the clause (we give the rounded value of ...)
14. Line 219: Again, this trend discussion needs a more rigorous error analysis if it is to be discussed, it is insufficient to just point to the standard error.
15. Line 232: Be specific. State which region has been associated with anthropogenic signals.
16. Line 256-259: Figure 2b suggests that there has been cooling in this sector of the transect. Is that cooling consistent with this circulation pattern and glacial changes?
17. Methods: Many of the trends observed in the section are much less than the temperature profile accuracy of 0.1C. This needs to be discussed.
18. Methods: Lines 326-333. I'm a little confused by the calculation of the CDW temperature and mean anomalies. The CDW is spread over a wide geographic region and covers a big depth range. So it could be that the 10% of warmest temperature grid points are not found in a cohesive region but rather spread far and wide. It might be useful to at least show what the horizontal and vertical spread is of these selected grid points is from cruise to cruise to see how these selected grid points behave.
19. Figure 1: Probably useful to label Hobart and Dumont d'Urville on this map, and also label the temperature contours within the section rather than call them out in the caption.
20. Figure S3: Perhaps overlay the standard deviation contours for each month so we can see where the variability in each month occurs?
21. Supplementary: Line 38: Should this be referring to Figure S3?
22. Supplementary: Line 56: Should this be referring to Figure S5?
23. Line 60: I'm confused why the standard deviation represents the gridding and measurement error estimation here. Doesn't this standard deviation represent the inherent temporal variability of the transect over the measurement period?
24. Figure S6: The STW "volume" has unusual mixed units – it would probably be better calculated as m^2 and obviously that is not a volume, but rather an area.
25. Figure S7: Discuss statistical significance (or not) of the trend in wind stress curl. Include the description of the gray shading in the caption.

Reviewer #2 (Remarks to the Author):

This manuscript is an interesting description of long term temperature changes in summer, along XBT Section IX28 in the Southern Ocean, over a period of 25 years. These changes are discussed in the context of interannual variability. Warming is observed for subantarctic waters and subsurface subpolar deep waters; cooling is observed in the near-surface subpolar waters. Warming of the subsurface subpolar deep waters is associated with a large shallowing (i.e. larger than estimated in previous studies). Also, the amplitude of the long-term temperature change is the lowest of the section, yet the signal-to-noise ratio is the largest because interannual variability is very weak in this sector.

The manuscript is generally well written.

Comments/Questions:

Line 286: In the supplemental material, it would be helpful to see a plot of the distance of available observations from the mean transect, i.e. x-axis: time (each profile is taken at a different time); y-

axis: distance from the southern most point of the mean transect; color: distance of available profile from the mean transect (positive or negative according to the location being to the east or west of the mean transect). Is there any structure of this distance in time at any location along the transect ? If so, how can it affect the analysis?

Line 288: Did you try the Piecewise Cubic Hermite Interpolating Polynomial (PCHIP) method for vertical interpolation? (instead of linear) How sensitive are the results to the interpolation method and to the resolution for the vertical grid selected?

How do you expect the data distribution (changing in time and space) to affect the trend and interannual variability estimate? Have you tried subsampling the data to test the robustness of the results? It would be valuable to include this discussion.

Suggested minor edits:

Fig. 1: I suggest to indicate in the caption what the different acronyms in the figure are.

Fig. 2: In panel C and the caption, I suggest to describe the numerator in the ratio using $Trend * \Delta_t$ and define Δ_t as the length of the record. The current description ($Trend * 25$) may be a bit confusing as the trend is in degC/decade in panel A. You could also spell out that the ratio has the long term change (over 25 years) as numerator.

Line 31-33: This sentence can be clarified on the lines of the later description "Counter-intuitively, it is in the Upper Circumpolar Deep Water layer, where the long-term change amplitude is the lowest of the section, that the signal-to-noise ratio is the largest because interannual variability is actually very weak"

Line 85: 25 years (since November 1992)

Line 90: ")" is missing at the end of the line

Line 112: "-" is a typo

Reviewer #3 (Remarks to the Author):

The paper presents a new 25-year dataset of repeated XBT transects between Tasmania and Antarctica that help shed light on the significant climate change in the Southern Ocean. The paper presents the dataset (mean state, trends and significance), distinguishing among the main ocean water masses, and discusses possible mechanisms leading to the multi-decadal changes. However, these discussions are mostly based on relevant scientific literature rather than analyses performed by the authors.

As such, considering the value of the dataset itself, the paper will be of certain interest for the climate community evaluating the global and regional climate change signals during the last decades. The paper is well written and there are only a few minor issues to be fixed/clarified in my opinion, which I list below.

L28: "radically" is quite subjective statement for an abstract

L60: "with the exception of changes in atmospheric large-scale circulation": not clear if you refer to the large number of in-situ atmospheric observations (questionable anyway for this long period) or to some physical mechanisms, please clarify.

L109 and elsewhere: the use of STD to indicate standard deviation and/or standard error (which are

the same in the authors' formulation) is misleading and needs to be unified. I also suggest to cross-validate the use of the signal-to-noise ratio with more statistically robust methods, such as the non-parametric Mann-Kendall trend test or similar. For instance the concept in L132-133 is confusing: the significance of the trend is tested in comparison of the natural variability, so the distinction between significance and SNR (L132-133) is not really correct in my opinion.

L168: "when isolating only data points cooler than 0°C": I don't see any physical meaning of doing that. It could statistically be sound taking as threshold a quartile of the T distribution, but still a bit subjective. I suggest removing it.

L203: Similarly, the removal of a supposed outlier without a robust timeseries quality control appears subjective. Suggest removing it: the 2-digit approximation will be 0.05 anyway.

L280: XBT correction strategies seem a bit out-of-date, compared e.g. to IQuOd approaches. There is a reason for that?

L307: "quite similar": please reformulate with a quantitative statement

L327: "10% temperature" again appears subjective. What is the impact of choosing 25%, or the punctual depth of the warmest gridpoint? In other words, how robust is this choice?

I also encourage the authors to release the anomaly gridded dataset (upon acceptance of their paper) that will be of help for the climate, modeling and reanalysis community. As far as I understand only uncorrected and corrected XBT profiles are publicly available.

Reviewer 1: please find below our point-by-point response to reviewer #1. We copied the reviewer's comments in black. Our response is in orange. Citations from the manuscript are indicated in orange italicized text, and line numbers correspond to line numbers in the new version of the manuscript.

This paper uses a 25-year time series of subsurface temperature measurements in the Australian sector of the Southern Ocean to distinguish long-term trends from year-to-year variability. The paper is very well written and the science is sound and represents a substantial contribution to the field. In particular, the results are important since we have very few long observational time series in the Southern Ocean and so much of our understanding of long-term changes in the region stem from models or sparse observational measurements of shorter duration. The unique aspect of this 25-year XBT transect is that the temperature profile data are collected along a near-repeat section and so it enables the analysis to be able to tease out temporal from spatial variability. I find that the paper should be suitable for publication in Nature Communications after addressing some of my concerns below.

We thank the reviewer for their careful reading, relevant comments, and enthusiastic support for our work. We hope that our responses below will convince the reviewer.

Major Issues:

1. The linear trends could use a more rigorous error analysis rather than just comparing to the standard error of the variability.

Accepted. Although, linear trends were compared to their standard error and interannual variability, we have now added a Mann-Kendall test for each trend computation. It allows us to further present the significance of each presented trend. Our results remain unchanged.

2. The authors use a 25-year time series but it mainly consists of measurements from the summer season (NDJF) that are then used to form estimates of annual temperature trends. The paper convincingly shows the long-term trends from the constructed “annual” trends in some regions lie above the year to year variability. But this assumes that there are no counteracting trends that might occur in winter-time measurements. This needs to be recognized and more fully discussed in the text.

Accepted. We have now added a note in the main text as well as clearer discussion in Methods, part c (see below). Although we do not include the figure in the paper, mostly for space constraints, we note here that the trends in surface temperature computed from only the summer months (November-February) or using all months of the year are essentially the same (Figure R1). We have now clarified that aspect in the text, as suggested by the reviewer, both in the main text and in the Methods section.

(Line 345-347) *“Surface trends are consistent with SST Reynolds⁵² product trends calculated over the summer NDJF period (Figure 2a., $r = 0.70$) but also for SST Reynolds⁵² full year trends ($r = 0.70$).”*

(Line 327-333) *“We note that even if measurements are sampled only in summertime, computed trends can be considered as annual trends. Indeed, the main seasonal variations are in the surface layer, and the XBT temperature profiles' surface values are consistent with satellite SST values. Finally, SST trends computed from NDJF months are coherent with SST trends computed over the full year. This shows that for the surface layer, there are no wintertime trends that are counteracting the summer trends, and observed trends are consistent for the whole year for the full time series.”*

Figure R1: Left: Temperature trends from SST Reynolds monthly anomalies over 1993-2017. Right: Temperature trends from summer (NDJF) SST Reynolds monthly anomalies over 1993-2017

3. Discussion of findings to other regions of the Southern Ocean?

Accepted. We have now added the Sprintall (2008) reference, which analyses similar long-term upper-ocean temperature changes in Drake Passage over 36 years. The author divides the transect into two zones, being north and south of the Polar Front. The subsurface trend north of the Polar Front was $\sim 0.02^{\circ}\text{C}/\text{year}$, consistent with our region A trend. A cooling of $-0.04^{\circ}\text{C}/\text{year}$ at the surface was found at the surface south of the PF, one order stronger than our region B, which was expected as the Reynolds SST cooling amplitude is stronger in the Drake Passage (Figure 2 a.).

(Line 255-258) “. The warming of $0.29 \pm 0.09^{\circ}\text{C}$ per decade north of the ACC is in accordance with previous studies⁷⁻⁹ and with other parts of the Southern Ocean⁴⁵. Southern Ocean circulation being essentially zonal, subsurface trends are expected to be zonally consistent all-around Antarctica.”

Sprintall, J. Long-term trends and interannual variability of temperature in Drake Passage. *Progress in Oceanography* 77, 316–330 (2008).

Minor Issues:

1. Abstract: lines 31-33. Why this result is “counter-intuitive” is not clear here. Obviously, it becomes clear later in the paper why this is an unexpected result but the abstract should stand alone. The sentence needs a qualifier to help the reader understand why the result is counter-intuitive. In addition, the clause “largest change of the section regarding interannual variability” is also a bit ambiguous. As I read it in your analysis, despite showing only small long-term trends, Region C has one of the most robust long-term trend along the section as the trend rises well above the year-to-year variability.

Accepted. We rephrase to: (Line 31-32) *“Although this subsurface warming of subpolar deep waters is small, it is the most robust long-term trend of our section, being in a region with weak interannual variability.”*

2. Abstract: line 36. West and East are a bit ambiguous when referring to a polar region. I realize that there is the Western Antarctic Peninsula, but the region that I think you are referring to is in the Bellingshausen Sea. Perhaps instead of West Antarctica you could specifically state that you are referring to the Bellingshausen Sea downstream from your study area.

Accepted. As suggested, we changed *“Western Antarctica”* to *“Amundsen – Bellingshausen Seas”*.

3. Lines 49-55. This is a long sentence that needs to be broken up for readability.

Accepted. We rewrote the sentence to (line 49-55) *“The overall water-mass warming since 1970 is composed of significant warming north of, and within, the eastward flowing Antarctic Circumpolar Current⁷⁻⁹ (ACC), and slight cooling observed in the surface subpolar waters¹⁰. Some regions show slight warming and uplifting of the subpolar Upper Circumpolar Deep Waters (that lie directly offshore the Antarctic continental shelf), threatening to invade onto the continental shelves with drastic potential consequences for the melt of Antarctic Ice Shelves and subsequent global sea level rise^{11”}.*

4. Lines 96-98: Figure 1 also distinguished LCDW at the far south along the transect. Should this also be discussed in this paragraph?

Accepted. We added a discussion of the LCDW as suggested by the reviewer: (line 95-99) *“Below the Winter Water tongue lies the less-dense Upper Circumpolar Deep Water (UCDW), then the denser Lower Circumpolar Deep Water (LCDW), that rises beneath the WW layer south of the Antarctic Divergence around 63°S. These Circumpolar Deep Waters are advected at depth around the Southern Ocean, and partly originate from North Atlantic Deep Waters^{25”}.*

5. Lines 96: As I have indicated above, a more rigorous error analysis probably is needed to help discern which trends are statistically significant. Although helpful, a standard error is not sufficient to statistically determine whether a linear trend is significant or not. At the very least the uncertainty of a linear trend at 95% confidence is related to twice the standard error. I suggest the authors use more standard statistical tests, such as t-tests that account for effective degrees of freedom to test for their significance of the linear trends.

Accepted. See our response above for details. A Mann-Kendall test have been added for all computed trends.

6. Line 118: Does a 25 year time series really resolve decadal time scales?

We agree with the reviewer that 25 years might not be enough to fully resolve decadal variability signal, though this is a typical timescale at which climate scale signal is diagnosed. To avoid misinterpretation, we have now removed the comment on decadal timescale.

7. Line 150: “Deep-reaching localized cells”. Rather than these features being called “cells” which suggests a circulation feature, could they be associated with shifting preferences in the location of the main frontal features that vary from cruise to cruise and year to year? This also points to the issue that averaging the temperature data by latitude necessarily convolves variability in the fronts into this geographical space. Ideally this averaging might be performed for different frontal regions. This probably needs to also be addressed in the paper.

We thank the reviewer for raising this important point. First, we agree that “cell” is confusing: we have removed this terminology. Regarding the impact of frontal meandering: we had carefully addressed that point in a previous study on this dataset. Our results show that the mean path of the fronts have minor impact on our results, but that eddy variability does explain most of the extrema in the norther region of the section, as stated in the text.

In Auger et al., (2019; doi: 10.5281/zenodo.4094960) we investigated the variability of the meanders of the fronts compared to the vessel trajectory, and demonstrated it had no significant impact on the computed trends. Figure R2 (reproduced here from Auger et al., 2019) shows the position of the main front of the region (polar front) at the time of the transect (plain colored lines), versus the transect of the ship (dotted line). The position of the front at the time of the transect is estimated from satellite altimetry. For clarity, we show here only the transect for the season 2007-2008. We are then able to extract a best estimate of the latitude of the front, with regard to either the exact transect or the mean transect. Doing this analysis for each transect of the full time series, for both the Subantarctic Front and the Polar Front, we are able to remove all transects where the intersection front/transect is substantially distant from the latitude of the intersection of mean transect/mean front. We found that removing or keeping these outlier transects did not induce any significant differences in our trend calculations, and did not change the conclusions of the study.

We now refer to this earlier study and have clarified our text: (Line 151-154) “*These structures may be related to more prevalent warm-core eddies or small meanders towards the end of the time series. We note that the warming trends are similar when analyzed in streamwise coordinates following altimetric-derived meanders or in geographical co-ordinates as presented here (Auger, 2019)*”.

Figure R2: Dotted lines: Path of each transect for season 2007-2008. Dashed black line is the mean transect. Solid lines are the position of the front the day of the transect derived from altimetry.

8. Line 159: What is the 0.64 correlation referring to? Between what time series?

We now clarified: (line 162-166) “*Similar interannual variability is also evident in the sea-surface temperature fields, with a correlation of 0.64 between SST and the Region A temperature time series, and a slightly lower 25-yr trend of $0.15 \pm 0.09^\circ\text{C}$ per decade, consistent with the trend distribution within the zone (Figure 2b).*”

9. Line 167: Should this refer to Figure 2c here rather than 3b?

Accepted. We changed the reference: (line 171-173) *“The overall cooling in the surface subpolar waters close to Antarctica, from the surface to 200 m and from 66°S to 61°S (region B), has a trend of $-0.07\pm 0.04^{\circ}\text{C}$ per decade (Fig. 3b, significant at 88%), with a signal-to-noise ratio of 1.16.”*

10. Line 175: Is this the correct Figure citation? What is the correlation between?

We apologize for the confusion. The reviewer is correct, we meant to refer to Figure 3b. We have corrected our text and also indicated the correlation, as requested by the reviewer:

(line 182-184) *“Such high-latitude cooling over the upper 200 m in region B is also consistent with local sea surface cooling observed from satellite SST observations (Figure 3b, correlation $r=0.80$)”*

11. Line 180-182: I wanted a bit more elaboration here about what the specific relationship is. What are the mechanisms that are being invoked for the freshening?

Accepted. We now elaborate a bit more on the underlying mechanism: (line 189-193) *“Indeed, a trend in surface water freshening has been observed over the same period near 140°E⁵. This has been linked to increased sea-ice cover, particularly after the Mertz Glacier calving in 2010 and enhanced by a large-scale northward shift of the zero-zonal wind position from 1999 onwards, that increased the Ekman-driven sea-ice convergence near the coast⁵”*

12. Lines 195-197: Reference Figure 3c here.

Accepted.

13. Line 204: I think you can delete the clause (we give the rounded value of ...)

Accepted, we now only give the rounded value as suggested.

14. Line 219: Again, this trend discussion needs a more rigorous error analysis if it is to be discussed, it is insufficient to just point to the standard error.

We thank the reviewer for the suggestion. We have now performed further analysis, which show that wind stress curl trends are not significant. We have now clarified that point in the paper: (line 227-232) *“The cause of the shallowing of the layer remains unclear. It could be related to long-term changes in Ekman pumping¹¹, but using the atmospheric reanalysis ERA-5, we find only a very subtle long-term trend in local upward Ekman pumping, which is not statistically significant. Other potential mechanisms, e.g., associated with turbulence-driven shallowing of the surface layer, remain to be tested in a future study.”*

15. Line 232: Be specific. State which region has been associated with anthropogenic signals.

Accepted. We revised the text to: (line 242-246) *“Interestingly, only region A has been shown to be associated with a human-induced forced signal that emerges over natural variability¹⁴, though recent work suggests that forced warming in the sub-surface subpolar ocean does emerge over natural variability by the end of the 20th century or early decade of the 21st⁴²”*

16. Line 256-259: Figure 2b suggests that there has been cooling in this sector of the transect. Is that cooling consistent with this circulation pattern and glacial changes?

The surface cooling might be consistent with glacial discharge, as explored in recent studies (Rye et al., 2020): glacial discharge would increase the surface stratification and cause surface cooling and subsurface warming. However, those processes are still debated, and we cannot enter in the details of

the processes at this stage of the paper. For clarity and avoid confusion we decided to remove that sentence from the paper.

Rye, C. D. et al. Antarctic Glacial Melt as a Driver of Recent Southern Ocean Climate Trends. *Geophysical Research Letters* 47, e2019GL086892 (2020).

17. Methods: Many of the trends observed in the section are much less than the temperature profile accuracy of 0.1C. This needs to be discussed.

Accepted. We have clarified that aspect in the Method section c: (line 341-345) *“Each grid point is sampled by between 3 and 10 profiles per year. With an XBT accuracy of 0.1°C, it translates into a standard error of the mean from ~0.03-0.06°C, allowing to resolve changes over 25 years of 0.001-0.002°C per year, or 0.01-0.02°C per decade. This value is lower when computing trends over larger regions A, B and C.”*

Number of profiles per year are shown figure R3.

Figure R3: Number of profiles per season for each grid point.

18. Methods: Lines 326-333. I'm a little confused by the calculation of the CDW temperature and mean anomalies. The CDW is spread over a wide geographic region and covers a big depth range. So it could be that the 10% of warmest temperature grid points are not found in a cohesive region but rather spread far and wide. It might be useful to at least show what the horizontal and vertical spread of these selected grid points is from cruise to cruise to see how these selected grid points behave.

We thank the reviewer for the suggestion. Figure R4 shows the 25 years evolution of the 10% warmest CDW points. It shows that most of warmest points are found in patches, or dipoles centered on latitude -57°S , which is consistent with the warming pattern Figure 2 b. There is no tendency in the presence of dipoles or patches found in the zone.

Figure R4: Position of the 10% warmest CDW points for each year.

19. Figure 1: Probably useful to label Hobart and Dumont d'Urville on this map, and also label the temperature contours within the section rather than call them out in the caption.

Accepted.

Figure 1 and caption modified.

20. Figure S3: Perhaps overlay the standard deviation contours for each month so we can see where the variability in each month occurs?

For clarity of the figure, we prefer not adding contours of standard deviation on Figure S3. Section of standard deviation within each month are dominated by large standard deviation in the northern part of the transect, where temperature changes are higher and deep reaching within a month due to the seasonal cycle. We tried to add contours as suggested by the reviewer, but the figure became overloaded and difficult to read, losing clarity on the main messages of this figure which is the temperature changes at the surface and the northern part of the trend along the Summer season.

21. Supplementary: Line 38: Should this be referring to Figure S3?

22. Supplementary: Line 56: Should this be referring to Figure S5?

Accepted. We apologize for the confusion. We adjusted the following references:

(Line 48 of supplementary) changed Figure S1 to Figure S2.

(Line 49 of supplementary) changed Figure S2 to Figure S3.

(Line 67 of supplementary) changed Figure S4a to Figure S5.

23. Line 60: I'm confused why the standard deviation represents the gridding and measurement error estimation here. Doesn't this standard deviation represent the inherent temporal variability of the transect over the measurement period?

Standard deviation is computed only over points sampled on the same grid point for one transect; ie: One profile is sampled at 58.8°S, another one is sampled at 58.9°S, both are interpolated on the 2m resolution grid, and associated with latitude 59°S. Resulting gridded temperature is the mean value of both profiles at each depth, so we can evaluate the gridding and measurement error by taking the standard deviation of all profile for one grid point.

24. Figure S6: The STW "volume" has unusual mixed units – it would probably be better calculated as m² and obviously that is not a volume, but rather an area.

Accepted. We computed STW area in m² in Figure S6 and changed to area in the label.

Line 168. "*impacting the extent of STW*"

Supplementary S6: All "*volume*" changed to "*extent*".

25. Figure S7: Discuss statistical significance (or not) of the trend in wind stress curl. Include the description of the gray shading in the caption.

Accepted. We now clarified that trends are not significant. As responded above, we also clarified the main text.

Reviewer 2: please find below our point-by-point response to reviewer #2. We copied the reviewer's comments in black. Our response is in orange. Citations from the manuscript are indicated in orange italicized text, and line numbers correspond to line numbers in the new version of the manuscript.

This manuscript is an interesting description of long term temperature changes in summer, along XBT Section IX28 in the Southern Ocean, over a period of 25 years. These changes are discussed in the context of interannual variability. Warming is observed for subantarctic waters and subsurface subpolar deep waters; cooling is observed in the near-surface subpolar waters. Warming of the subsurface subpolar deep waters is associated with a large shallowing (i.e. larger than estimated in previous studies). Also, the amplitude of the long-term temperature change is the lowest of the section, yet the signal-to-noise ratio is the largest because interannual variability is very weak in this sector.

The manuscript is generally well written.

We thank the reviewer for their careful reading and relevant comments.

Comments/Questions:

Line 286: In the supplemental material, it would be helpful to see a plot of the distance of available observations from the mean transect, i.e. x-axis: time (each profile is taken at a different time); y-axis: distance from the southern most point of the mean transect; color: distance of available profile from the mean transect (positive or negative according to the location being to the east or west of the mean transect). Is there any structure of this distance in time at any location along the transect ? If so, how can it affect the analysis?

We thank the reviewer for the suggestion. We did produce the figure suggested by the reviewer (Figure R5). Figure R5 show no long-term pattern emerging that could explain our results, and trends in distance from the mean path are weak. The strongest trend is 12 km per decade, representing a 30 km drift along the 25 years, and none of them is significant with a Mann-Kendall test. We do not expect this kind of bias to affect the trends.

Impact of variations of sampling distance from the mean transect has been studied in preliminary studies, by removing transects impacted by variation of position of the SubAntarctic and Polar fronts.

We have now clarified that aspect:

(Line 151-154) *“These structures may be related to more prevalent warm-core eddies or small meanders towards the end of the time series. We note that the computed warming is similar when analyzed in streamwise co-ordinates following altimetric-derived meanders or in geographical co-ordinates like presented here²⁶”.*

Figure R5: Top panel: Distance of available observations from the mean transect. Position of the dots is dictated by distance from the Southernmost point of the mean transect, and time, and color represents the distance from the mean transect. Bottom panel: Trend in distance from mean path for each 500km step along the mean path. Error bars are the standard errors of each trend.

Line 288: Did you try the Piecewise Cubic Hermite Interpolating Polynomial (PCHIP) method for vertical interpolation? (instead of linear) How sensitive are the results to the interpolation method and to the resolution for the vertical grid selected?

The interpolation is strongly constrained by observations, so our results are insensitive to the choice of the vertical interpolation. Depth resolution of the interpolation is three times greater than the sampling of the sampled profile (resolution of the original profile is less than one meter, 0.66 m; resolution of the interpolated profile is 2 m).

We nevertheless investigated the sensitivity by computing cubic vertical interpolation. By doing so and re-performing the entire analysis, Figures 2b and 2c show essentially the exact same pattern, both qualitatively and quantitatively. Changing the resolution of the vertical grid to 4m change the trend values up to 0.01°C per decade and doesn't change the conclusions.

We clarified that aspect in the Method section: (line 301-302): “Results are robust when changing the vertical resolution and vertical interpolation.”.

How do you expect the data distribution (changing in time and space) to affect the trend and interannual variability estimate? Have you tried subsampling the data to test the robustness of the

results? It would be valuable to include this discussion.

Effects of the data distribution changing in time are efficiently corrected by using anomalies computed with a daily gridded climatology. This helps us to consider potential biases emerging from samples taken at various time in the season or within a month. No interpolation has been used to fill the gaps within a transect for our trend calculations, to limit biases due to changes in spatial time distribution.

In addition, one part of our analysis investigates trends within larger geographical domains (region A-C; see Figure 2b and 2c), in which small changes in the data distribution would have virtually no effect.

As mentioned in the paper, we are also re-assured that our results are qualitatively consistent with past studies and quantitatively consistent with SST trends, despite the variability of the data distribution.

Subsampling to use only half of the profiles leads to similar but noisier trend estimates for Figure 2 b., and same trends for regions A, B and C:

Region A: $0.28 \pm 0.09^\circ\text{C}/\text{dec}$

Region B: $-0.07 \pm 0.04^\circ\text{C}/\text{dec}$

Region C: $0.04 \pm 0.01^\circ\text{C}/\text{dec}$

We have clarified the impact of data distribution in the Method section: (line 337-339): “*No interpolation was made in latitude to avoid interpolation of anomalies over large data gaps (eg during storms), so trends are robust to varying data distribution.*”.

Suggested minor edits:

Fig. 1: I suggest to indicate in the caption what the different acronyms in the figure are.

Accepted. We clarified acronyms in the caption: “*Figure 1: a) SURVOSTRAL observations over 25 years between Hobart and Dumont D’Urville (DDU). The mean trajectory is in dashed black. Data used in this study are in grey. A schematic circulation is represented. White, black and red arrows are respectively the Antarctic Circumpolar Current, the Antarctic Slope Current and Australian-Antarctic Basin gyre, and the East Australian Current. b) 25 year average of the summer (NDJF) mean temperature section. Average position of the fronts (SB: Southern Boundary, S-SACCF (Southern Branch of the Southern Antarctic Circumpolar Current Front, N-SACCF: Northern Branch of the Southern Antarctic Circumpolar Current Front, PF-S and PF-N are the Southern and Northern branches of the Polar Front, SAF: SubAntarctic Front, STF: SubTropical Front) and water-masses positions are indicated (LCDW: Lower Circumpolar Deep Water, UCDW: Upper Circumpolar Deep Water, AASW: Antarctic Surface Water, SAMW: SubAntarctic Modal Water, STW: SubTropical Water). Black contours show the (0° , 2° , 5° , 8° , 11°C) mean isotherms.*”

Fig. 2: In panel C and the caption, I suggest to describe the numerator in the ratio using $\text{Trend} \times \Delta t$ and define Δt as the length of the record. The current description ($\text{Trend} \times 25$) may be a bit confusing as the trend is in $\text{degC}/\text{decade}$ in panel A. You could also spell out that the ratio has the long term change (over 25 years) as numerator.

Accepted. We have clarified the caption as suggested by the reviewer: “*b) Temperature trends from SURVOSTRAL XBT data. Hatched data represent zones where $\text{abs}(\text{Trends} \times \Delta T) / \text{STD} < 1$, (here ΔT is the length of the record); i.e. where the trends are smaller than the interannual variability over the 25 years of measurements. c) is the ratio between the trend signal and interannual variability.*”

Line 31-33: This sentence can be clarified on the lines of the later description “Counter-intuitively, it is in the Upper Circumpolar Deep Water layer, where the long-term change amplitude is the lowest of the section, that the signal-to-noise ratio is the largest because interannual variability is actually very

weak"

Accepted. We clarified the sentence to (also addressing other comment from reviewer #1): (line 32-33) *“Although this subsurface warming of subpolar deep waters is small, it is the most robust long-term trend of our section, being in a region with weak interannual variability.”*

Line 85: 25 years (since November 1992)

Accepted.

Line 90: “)” is missing at the end of the line

Accepted. Bracket added.

Line 112: “-“ is a typo

Accepted.

Reviewer 3: please find below our point-by-point response to reviewer #3. We copied the reviewer's comments in black. Our response is in orange. Citations from the manuscript are indicated in orange italicized text, and line numbers correspond to line numbers in the new version of the manuscript.

The paper presents a new 25-year dataset of repeated XBT transects between Tasmania and Antarctica that help shed light on the significant climate change in the Southern Ocean.

The paper presents the dataset (mean state, trends and significance), distinguishing among the main ocean water masses, and discusses possible mechanisms leading to the multi-decadal changes. However, these discussions are mostly based on relevant scientific literature rather than analyses performed by the authors.

As such, considering the value of the dataset itself, the paper will be of certain interest for the climate community evaluating the global and regional climate change signals during the last decades. The paper is well written and there are only a few minor issues to be fixed/clarified in my opinion, which I list below.

We thank the reviewer for their careful reading, relevant comments, and support for our work. We have addressed all the points raised by the reviewer and we hope our responses below will convince the reviewer.

L28: "radically" is quite subjective statement for an abstract

Accepted. We changed the sentence to: (line 28-29) *“Three regions stand out as having strong trends that dominate over interannual variability”*

L60: "with the exception of changes in atmospheric large-scale circulation": not clear if you refer to the large number of in-situ atmospheric observations (questionable anyway for this long period) or to some physical mechanisms, please clarify.

We were referring to the results of Jones et al. (2016), which shows that the change of Southern Hemisphere westerly wind systems is larger than the typical natural variability recovered from long-term observation or paleo archives. But we agree with the reviewer that the mention was unclear, and is unnecessary at this stage in the paper. We chose to remove it from the sentence (line 59-61): *“For most changes in the Southern Hemisphere, it remains unclear whether the natural and interannual variability can cause the observed change or overwhelms the forced response”*

L109 and elsewhere: the use of STD to indicate standard deviation and/or standard error (which are the same in the authors' formulation) is misleading and needs to be unified. I also suggest to cross-validate the use of the signal-to-noise ratio with more statistically robust methods, such as the non-parametric Mann-Kendall trend test or similar. For instance the concept in L132-133 is confusing: the significance of the trend is tested in comparison of the natural variability, so the distinction between significance and SNR (L132-133) is not really correct in my opinion.

We thank the reviewer for this comment. We have now clarified what we mean by significant, and reserved the usage of significant to its statistical meaning. As suggested by the reviewer, we now use a non-parametric Mann-Kendall trend test, to test the significance of all of our trends. Our results remain unchanged. The statistical significance is therefore tested using a Mann-Kendall test; the confidence interval of the trends is computed as its standard error; and the standard deviation is used as an estimate of interannual variability, to estimate whether the computed 25-year change is larger than interannual variability.

We clarified that aspect in line 343-345: *“The trend significance is computed using a Mann Kendall test. Trends with p-value lower than 0.05 are considered significant, and their confidence interval is computed as their standard error.”* And line 361 *“The amplitude of the trend compared to the strength of the interannual variability is evaluated for each zone and grid point, by computing the signal to noise ratio.”*, and changed the following sentence:

(line 107-110) *“a cooling trend of 0.1 to 0.3°C per decade is observed in the coolest water-mass of the region (region B in Fig. 2b), extending from the surface to about 200 m, in a region where the interannual variability has similar magnitude”*

(line 111) *“but the trends are dominated by interannual variability”*

(line 113-114) *“subtle warming trends of around 0.05°C per decade are observed from -113 62.5°S to 52°S, but here, the interannual variability is weak.”*

(line 125-127) *“The three regions highlighted above clearly stand out, experiencing temperature changes that emerge above the background interannual variability over the past 25 years.”*

L168: "when isolating only data points cooler than 0°C": I don't see any physical meaning of doing that. It could statistically be sound taking as threshold a quartile of the T distribution, but still a bit subjective. I suggest removing it.

We respectfully disagree with the reviewer. Water-masses are defined by their hydrological properties, and here, while this cut-off isotherm is subjective, we investigate changes within the water-mass defined by waters cooler than 0°C. Figure 1 and 2, both show this water-mass as standing out at the southern edge of our section, with consistent long-term change. We here, want to highlight long term change within this water-mass.

We clarified this in the paper:

Line (174-178): *Figures 1b and 2bc both show water-mass cooler than 0°C as standing out at the southern edge of our section, with consistent long-term change. When subjectively isolating only data points cooler than 0°C, the cooling is significant and slightly more marked (-0.09±0.05°C per decade, signal-to-noise ratio of 1.49; Fig. 4a).*

L203: Similarly, the removal of a supposed outlier without a robust timeseries quality control appears subjective. Suggest removing it: the 2-digit approximation will be 0.05 anyway.

We agree our choice is entirely subjective. This is why our main conclusions and results are drawn entirely with the entire time-series. However, we find useful to discuss the fact that most of the interannual variability is due to only one outlier, so that we remove this single point, the signal to noise ratio increase drastically from 2.44 to 3.79. It is also useful to show that the trend is not induced by this single outlier point. This is purely used as a sensitivity test.

L280: XBT correction strategies seem a bit out-of-date, compared e.g. to IQuOd approaches. There is a reason for that?

We thank the reviewer to pointing to this important question of XBT corrections. In the previous version of our manuscript we were indeed using the official UNESCO correction based on the Hanawa fall rate correction. We have now entirely revised that strategy, and re-processed all of the correction and analysis using the most up-to-date correction. As part of this process, we used the help from Rebecca Cowley, who accordingly joined our team as a co-author of the paper. Based on her advice, we decided to use the Cheng et al 2014 correction which is the correction recommended the XBT Science group. All our results remain similar whether we use the Hanawa correction or the Cheng et al 2014 correction. This reprocessing allowed us to show the stability and robustness of our results. An

explication of the lack of differences due to the fall-rate corrections in this Southern Ocean region over the last 25-years has been added to Supplementary Materials S1.

Cheng, L., Zhu, J., Cowley, R., Boyer, T. & Wijffels, S. Time, Probe Type, and Temperature Variable Bias Corrections to Historical Expendable Bathythermograph Observations. *J. Atmos. Oceanic Technol.* 31, 1793–1825 (2014).

Hanawa, K., Rual, P., Bailey, R., Sy, A. & Szabados, M. A new depth-time equation for Sippican or TSK T-7, T-6 and T-4 expendable bathythermographs (XBT). *Deep Sea Research Part I: Oceanographic Research Papers* 42, 1423–1451 (1995).

Information (NCEI), N. C. for E. International Quality Controlled Ocean Database (IQuOD) version 0.1 - aggregated and community quality controlled ocean profile data 1772-2018 (NCEI Accession 0170893). <https://data.nodc.noaa.gov/cgi-bin/iso?id=gov.noaa.nodc:0170893#>.

L307: "quite similar": please reformulate with a quantitative statement

Accepted. We changed to “consistent with”.

L327: "10% temperature" again appears subjective. What is the impact of choosing 25%, or the punctual depth of the warmest gridpoint? In other words, how robust is this choice?

We agree our choice is subjective, and we have therefore performed a robustness analysis as suggested by the reviewer. 10% is a compromise between taking too many points and therefore underestimating the trend and taking too few points so being sensitive to isolated profiles that may have been taken in eddies.

In Figure R6, we estimated the depth (panel a) and temperature (panel b) trend, for a range of different percentile choices, from 0 to 50%. The red shading indicates the confidence interval of the trend (\pm one standard error). For depth trend, the magnitude of the trend is not weakly sensitive to the percentile taken, for percentile between 1 and 20%. The trend is more affected for larger percentile, but they seem unjustified to us. For temperature, trends peak at $0.07^{\circ}\text{C}/\text{dec}$ when taking 5 % of the warmest CDW points and is of the order of $0.05\text{-}0.06^{\circ}\text{C}/\text{dec}$ for 1, 10, 15, 20%, which again does not change our conclusions.

Figure R6: Top Panel: Depth Trends (m/dec) depending on the percentile of warmest point within CDW taken. Bottom Panel: Temperature Trends ($^{\circ}\text{C}/\text{dec}$) depending on the percentile of warmest point within CDW taken.

I also encourage the authors to release the anomaly gridded dataset (upon acceptance of their paper) that will be of help for the climate, modeling and reanalysis community. As far as I understand only uncorrected and corrected XBT profiles are publicly available.

Accepted. Previous version of our gridded product has been made available publicly on SEDOO: <https://doi.org/10.6096/11>. Update of the product with new XBT corrections will be done upon accepted manuscript, and the information with doi and links will be made available in the acknowledgment (or any more suitable section that the editor would indicate to us).

REVIEWERS' COMMENTS

Reviewer #1 (Remarks to the Author):

Review of "Southern Ocean in-situ temperature trends over 25 years emerge from interannual variability"

By Matthis Auger, Rosemary Morrow, Elodie Kestenare, Jean-Baptiste Sallée, Rebecca Cowley

November 2020

The authors have done a great job of addressing all my comments. I have only a few very minor suggestions for clarification.

Abstract: "This robust warming is associated with a large shallowing (39 ± 09 m per decade), which has been significantly underestimated by a factor of 3 to 10 in past studies."

Shallowing should be shoaling. But more so – a shoaling of what? Depth of the core maximum temperature in UCDW?

Line 112: A bit awkward. Suggest "Deeper in the water column, the Upper Circumpolar Deep Water layer (region C in Fig. 2b) shows subtle warming trends of ..."

Line 121: Suggest "(referred to here as noise)"

Line 225: "shallow" should be "shoal"

Line 227: "shallowing" should read "shoaling"

Line 263: "shallowing" should read "shoaling"

Line 272: "shallowing" should read "shoaling"

Reviewer #2 (Remarks to the Author):

The authors addressed all of my suggestions and questions and I have no further comments.

Reviewer #3 (Remarks to the Author):

This is the second review of the manuscript "Southern Ocean in-situ temperature trends over 25 years emerge

2 from interannual variability". The authors have completely and very satisfactorily addressed my previous concerns, in particular improving the uncertainty and significance analysis, updating the XBT fall rate correction method to a state-of-the-art one, and removing subjective statements throughout the manuscript.

I do not have further comments/concerns and I therefore recommend the manuscript for publication.

We thank all the reviewers for their comments and requests that helped to improve the manuscript. Here are the response to the remaining suggestion emitted by Reviewer#1.

Reviewer #1 (Remarks to the Author):

Review of "Southern Ocean in-situ temperature trends over 25 years emerge from interannual variability"

By Matthis Auger, Rosemary Morrow, Elodie Kestenare, Jean-Baptiste Sallée, Rebecca Cowley

November 2020

The authors have done a great job of addressing all my comments. I have only a few very minor suggestions for clarification.

Abstract: "This robust warming is associated with a large shallowing (39 ± 09 m per decade), which has been significantly underestimated by a factor of 3 to 10 in past studies."

Shallowing should be shoaling. But more so – a shoaling of what? Depth of the core maximum temperature in UCDW?

Accepted.

We changed: "This robust warming is associated with a large shallowing (39 ± 09 m per decade)" to: (Lines 34-35) *This robust warming is associated with a large shoaling of the maximum temperature core in the subpolar deep water (39 ± 09 m per decade)*

Line 112: A bit awkward. Suggest "Deeper in the water column, the Upper Circumpolar Deep Water layer (region C in Fig. 2b) shows subtle warming trends of ..."

Accepted.

We changed: "At deeper depth, in the Upper Circumpolar Deep Water layer (region C in Fig. 2b), subtle warming trends of around 0.05°C per decade are observed from -62.5°S to 52°S " to:

(Lines 112-114) *"Deeper in the water column, the Upper Circumpolar Deep Water layer (region C in Fig. 2b) shows subtle warming trends of around 0.05°C per decade from 62.5°S to 52°S "*

Line 121: Suggest "(referred to here as noise)"

Accepted.

We changed "referred to as noise" to (Line 121): "referred to here as noise", and "referred to as signal" to (Line 122): "referred to here as signal"

Line 225: "shallow" should be "shoal"

Line 227: "shallowing" should read "shoaling"

Line 263: "shallowing" should read "shoaling"

Line 272: "shallowing" should read "shoaling"

Accepted. Shallow changed to shoaling everywhere in the manuscript.